# Working memory facilitates reward-modulated Hebbian learning in recurrent neural networks

**Roman Pogodin**[1,*]**, Dane Corneil**[2]**, Alexander Seeholzer**[2]**, Joseph Heng**[2]**, Wulfram Gerstner**[2]
[1]Gatsby Computational Neuroscience Unit, University College London, London, UK
[2] School of Computer and Communication Sciences and School of Life Sciences,
Brain Mind Institute, École Polytechnique Fédérale de Lausanne, Lausanne, Switzerland
*roman.pogodin.17@ucl.ac.uk

## Abstract

Reservoir computing is a powerful tool to explain how the brain learns temporal sequences, such as movements, but existing learning schemes are either biologically implausible or too inefficient to explain animal performance. We show that a network can learn complicated sequences with a reward-modulated Hebbian learning rule if the network of reservoir neurons is combined with a second network that serves as a dynamic working memory and provides a spatio-temporal backbone signal to the reservoir. In combination with the working memory, reward-modulated Hebbian learning of the readout neurons performs as well as FORCE learning, but with the advantage of a biologically plausible interpretation of both the learning rule and the learning paradigm.

## 1 Introduction

Learning complex temporal sequences that extend over a few seconds – such as a movement to grab a bottle or to write a number on the blackboard – looks easy to us but is challenging for computational brain models. A common framework for learning temporal sequences is reservoir computing (alternatively called liquid computing or echo-state networks) [1, 2, 3]. It combines a reservoir, a recurrent network of rate units with strong, but random connections [4], with a linear readout that feeds back to the reservoir. Training of the readout weights with FORCE, a recursive least-squares estimator [1], leads to excellent performance on many tasks such as motor movements.

The FORCE rule is, however, biologically implausible: update steps of synapses are rapid and large, and require an immediate and precisely timed feedback signal. A more realistic alternative to FORCE is the family of reward-modulated Hebbian learning rules [5, 6, 7], but plausibility comes at a price: when the feedback (reward minus expected reward) is given only after a long delay, reward-modulated Hebbian plasticity is not powerful enough to learn complex tasks.

Here we combine the reservoir network with a second, more structured network that stores and updates a two-dimension continuous variable as a "bump" in an attractor [8, 9]. The activity of the attractor network acts as a dynamic working memory and serves as input to the reservoir network (fig. 1). Our approach is related to that of feeding an abstract oscillatory input [10] or a "temporal backbone signal" [11] into the reservoir in order to overcome structural weaknesses of reservoir computing that arise if large time spans need to be covered.

In computational experiments, we show that a dynamic working memory that serves as an input to a reservoir network facilitates reward-modulated Hebbian learning in multiple ways: it makes a biologically plausible three-factor rule as efficient as FORCE; it admits a delay in the feedback signal; and it allows a single reservoir network to learn and perform multiple tasks.

NeurIPS 2019 workshop "Real Neurons & Hidden Units: Future directions at the intersection of neuroscience and artificial intelligence", Vancouver, Canada.

## 2 Model

Our architecture is simple: the attractor network (the "memory") receives some task-specific input and produces a robust two-dimensional neural trajectory; the reservoir network (the "motor cortex") shapes its dynamics with this trajectory, and produces a potentially high-dimensional output (fig. 1).

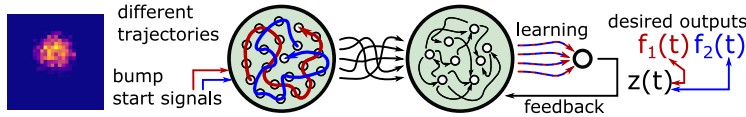

Figure 1: Model architecture: a moving 2D bump (left: activity bump (red) surrounded by inactive neurons (blue)) in an attractor network (left circle with two bump trajectories) projects to a reservoir (right circle); the output $\mathbf{z}(t)$ is read out from the reservoir and approximates the target function.

**Attractor network.** Following [9], the bump attractor consists of 2500 neurons evolving as

$$
\begin{aligned}
\tau_{\mathrm{m}}\dot{\mathbf{x}} &= -\mathbf{x} + [\mathbf{J}\mathbf{x} + \mathbf{e} - \mathbf{h}]_+ \,, \\
\tau_{\mathrm{a}}\dot{\mathbf{h}} &= -\mathbf{h} + s\,\mathbf{x}\,,
\end{aligned}
\tag{1}
$$

where $\mathbf{x}$ is the vector of firing rates evolving with time constant $\tau_{\mathrm{m}}$, $\mathbf{e}$ is the task-specific external input, $\mathbf{h}$ is an adaptation variable with time constant $\tau_{\mathrm{a}}$ and $s$ is the strength of adaptation. The weight matrix $\mathbf{J} = \mathbf{J}_{\mathrm{s}} + \mathbf{J}_{\mathrm{h}}$ has two parts. The symmetric part $\mathbf{J}_{\mathrm{s}}$ creates a two-dimensional translation-invariant structure resulting in bump-like stable activity patterns, whereas $\mathbf{J}_{\mathrm{h}}$ represents structural noise. Due to the adaptation $\mathbf{h}$, the bump moves across a path defined by the initial conditions and structural noise, creating long–lasting reliable activity patterns which also depend on the input $\mathbf{e}$.

**Reservoir network.** The reservoir learns to approximate a target function $\mathbf{f}(t)$ with the output $\mathbf{z}(t)$ by linearly combining the firing rate $\mathbf{r}$ with readout weights $\mathbf{W}_{\mathrm{ro}}$: $\mathbf{z} = \mathbf{W}_{\mathrm{ro}}\mathbf{r} + \boldsymbol{\eta} \equiv \hat{\mathbf{z}} + \boldsymbol{\eta}$ with readout noise $\boldsymbol{\eta}$. We use the same number of neurons (1000) and parameters as [1, 6],

$$
\tau\dot{\mathbf{u}} = -\mathbf{u} + \lambda\mathbf{W}_{\mathrm{rec}}\mathbf{r} + \mathbf{W}_{\mathrm{fb}}\mathbf{z} + c\,\mathbf{W}_{\mathrm{attr}}\mathbf{x}\,, \quad \mathbf{r} = \tanh(\mathbf{u}) + \boldsymbol{\xi}\,,
\tag{2}
$$

where $\mathbf{u}$ is the membrane potential, $\boldsymbol{\xi}$ is the firing rate noise, $\mathbf{W}_{\mathrm{attr}}$ scales attractor input with coupling $c$, $\mathbf{W}_{\mathrm{rec}}$ and $\lambda$ regulate chaotic activity [4], and $\mathbf{W}_{\mathrm{fb}}$ implements the feedback loop.

**Learning rule.** We use the reward-modulated Hebbian rule of [6] for the readout weights $\mathbf{W}_{\mathrm{ro}}$,

$$
\Delta\mathbf{W}_{\mathrm{ro}}(t) = \eta(t)M(t)(\mathbf{z}(t) - \bar{\mathbf{z}}(t))\mathbf{r}^\top(t)\,, \quad \eta(t) = \eta_0/(1 + t/\tau_\eta)\,,
\tag{3}
$$

where $\bar{x}$ denotes low-pass filtering of $x$, such that $\mathbf{z}(t) - \bar{\mathbf{z}}(t) \approx \boldsymbol{\eta}(t)$. The reward modulation $M(t)$ tracks performance $P(t)$ as

$$
P(t) = -\|\mathbf{f}(t) - \mathbf{z}(t)\|^2\,, \quad M(t) = \begin{cases} 1, & P(t) > \bar{P}(t)\,, \\ 0, & P(t) \le \bar{P}(t)\,. \end{cases}
\tag{4}
$$

The update rule is an example of a NeoHebbian three-factor learning rule [12] and mimics gradient descent if we ignore the feedback loop [5]. For model details, see appendix A.

## 3 Experiments

In fig. 2, the learning rules are compared on 50 target functions sampled from a Gaussian Process (GP) with exponential squared kernel ($\sigma^2 = 10^4$ to match the complexity of hand-picked functions from [1, 6]). After each training period, we measure performance with normalized cross-correlation between the output and the target (ranging from -1 to 1, where 1 is a perfect match) on a single trial with frozen weights. Details are provided in appendix A; **code**: https://github.com/neuroai-workshop-anon-1224113/working-memory-facilitating-reservoir-learning.

### 3.1 Reward-modulated Hebbian learning with attractor input reaches FORCE performance

When tested on one-second signals similar to those of [1, 6] (two insets in fig. 2), the full network with attractor input and reward-modulated Hebbian learning learns faster and more reliably than

reward-modulated Hebbian learning without the input from the attractor network. After about 90 training trials, the full network achieves the performance of the FORCE rule (for which training error approaches one in the first trial, [1], while test error does so after 30-50 trials, [6]; fig. 2A).

For target signals that extend over 10 seconds (same smoothness of the target functions, two insets in fig. 2B), the reward-modulated Hebbian rule achieves a performance of 1 after 200 trials if combined with input from the attractor network (fig. 2B) but fails completely without the attractor network (tuning of the hyperparameters on a logarithmic scale did not help; data not shown). Thus a three-factor learning rule succeeds to learn complex tasks if combined with a temporally structured input from the attractor network.

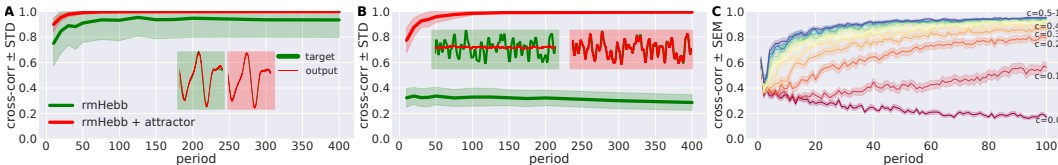

Figure 2: **A.** Mean test performance as a function of trials with a reward-modulated Hebbian (rmHebb) learning rule combined with (red) or without (green) input from attractor. Inset: target function over 1s and examples of the output after training. **B.** Same, but for a target signal that spans over 10s. Shaded area shows standard deviation. The FORCE rule achieves a performance of 1 in all trials. **C.** Learning performance for different couplings $c$ of the attractor output to reservoir input ($c = 1$ normal coupling, $c = 0$ no coupling) when updates are delayed to the end of every one-second trial.

## 3.2 Attractor input allows realistic frequency of weight updates

FORCE learning needs a feedback signal at every time step. Standard reward-modulated Hebbian learning can support very small delays, but fails if updates are less frequent than every few ms [6]. In our approach (fig. 2C), proposed updates are summed up in the background, but applied only at the end of a one-second trial. We find that even with such a temporally sparse update, learning is still possible.

The input from the dynamic working memory is necessary to achieve this task: when the strength of the input from the attractor network gradually decreases, performance drops; in the total absence of attractor input ($c = 0.0$; note that the reservoir still receives weak input noise) learning completely fails. Strikingly, delayed updates do not hurt performance, and the system achieves high ($> 0.9$) cross-correlation in fewer than 100 training trials if the input from the attractor network is strong enough. The transient drop in performance shortly after the start in fig. 2C is likely due to $\mathbf{W}_{\text{ro}} = 0$ in the beginning, meaning that the output is uncorrelated with the firing rates, and therefore the cumulative weight update does not approximate gradient information.

## 3.3 Working memory translates to efficient reservoir learning of multiple signals

It is well known that reservoir networks can learn multiple tasks given different initial conditions with both FORCE [1] and the reward-modulated Hebbian rule [6]. We want to check whether this also holds for our approach. We conjecture that different inputs to the attractor network generate unique neural trajectories [9] that can be exploited by the reservoir network.

To test this hypothesis, we train the network to produce hand-written digits. The **static** input to the attractor comes from the pre-processed MNIST dataset (network inputs are taken from one of the last layers of a deep network trained to classify MNIST) in order to provide a realistic input to the attractor network which transforms the static input into dynamic trajectories (noiseless, fig. 3B, and noisy, fig. 3D). We record 50 attractor trajectories used for training (used 4 times each, resulting in 2000 training trials) and 50 for testing of each digit (1 second each), where each trajectory corresponds to a distinct input pattern. The reservoir learns a single drawing for each class. The variance of the structural noise in the attractor network is 3 times larger compared to the previous experiments in order to produce more robust bump trajectories (fig. 3D).

The reward-modulated Hebbian rule masters 10 out of 10 digits when driven by a noiseless input from the attractor network (fig. 3A). In the presence of noise in the attractor network (fig. 3D), the

performance is imperfect for "five" and "six" (fig. 3C). We checked that FORCE learning with the same noisy input did not improve the performance (data not shown). Note that a linear readout of the attractor (without the reservoir) would be insufficient: first, sometimes single digit trajectories are very dissimilar (e.g. the different zero's in fig. 3D); second, at points where trajectories cross each other, a delay-less linear readout must produce the same output, no matter what the digit is.

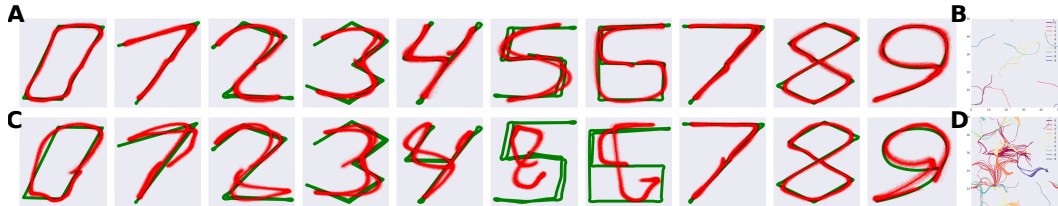

Figure 3: **A.** Targets (green) and test outputs (red) for reward-modulated Hebbian rule with noiseless attractor input. **B.** Noiseless attractor trajectories for each digit. **C.** Same as **A** but with noisy attractor trajectories. **D.** Same as **B** but static input's noise resulted in noisy trajectories (individual lines).

## 4    Discussion

We showed that a dynamic working memory can facilitate learning of complex tasks with biologically plausible three-factor learning rules. Our results indicate that, when combined with a bump attractor, reservoir computing with reward-modulated learning can be as efficient as FORCE [1], a widely used but biologically unrealistic rule. The proposed network relies on a limited number of trajectories in the attractor network. To increase its capacity, a possible future direction would be to combine input from the attractor network with another, also input-specific, but transient input that would bring the reservoir into a different initial state. In this case the attractor network would work as a time variable (as in [9]), and the other input as the control signal (as in [1]).

Apart from the biological relevance, the proposed method might be used for real-world applications of reservoir computing (e.g. wind forecasting [13]) as it is computationally less expensive than FORCE. It might also be an interesting alternative for learning in neuromorphic devices.

### Acknowledgments

This research was supported by the Gatsby Charitable Foundation, Swiss National Science Foundation (no. 200020 - 184615) and by the European Union Horizon 2020 Framework Program under grant agreement no. 785907 (HumanBrain Project, SGA2). The authors thank Moritz Deger for an earlier version of the reservoir code and Jorge Aurelio Menendez for useful comments on the manuscript.

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

# A  Supplementary Materials

**Simulation details.** Both networks were simulated with the Euler method with the step size $dt = 1$ ms. The attractor network dynamics was recorded after a 100 ms warm up period to allow creation of the bump solution, during which it received additional input from the images in section 3.3. Training was done consequently, without breaks in the dynamics between trials. For testing, the network started from the preceding training state and continued with frozen weights. After testing, the pre-training activity was restored. The code for experiments is available at https://github.com/neuroai-workshop-anon-1224113/working-memory-facilitating-reservoir-learning.

**Test functions.** Gaussian process test function were drawn from
$$f_{GP} \sim \mathcal{GP}(0, K), \ K(x, y) = \exp\left(-(x - y)^2/(2\sigma^2)\right). \tag{5}$$
Forcing both ends of the function to be zero and denoting $\mathbf{x} = (0, T - 1)^\top$, $\mathbf{z} = (1, \ldots, T - 2)^\top$, we sample test functions as
$$f_{GP}(1, \ldots, T - 2) \sim \mathcal{N}\left(\mathbf{0}, K(\mathbf{z}, \mathbf{x})K^{-1}(\mathbf{x}, \mathbf{x})K(\mathbf{x}, \mathbf{z})\right), \tag{6}$$
where $T$ is either $10^3$ (short tasks) or $10^4$ (long tasks). We chose $\sigma^2$ to roughly match the complexity of targets from [6] ($\sigma^2 = 10^4$). 50 random functions were tested on 50 random reservoirs that nevertheless received the same attractor input ($\mathbf{W}_{\text{attr}}$ was not resampled). In section 3.3, the same reservoir was used for all runs. The noisy input for section 3.3 was taken from an intermediate layer of a deep network trained to classify MNIST, and the noiseless input stimulated only a 5 by 5 square of neurons (unique for each digit).

**Attractor network parameters.** The time constants were $\tau_{\text{m}} = 30$ ms, $\tau_{\text{a}} = 400$ ms. Adaptation strength was $s = 1.5$. The external input $\mathbf{e}$ was drawn independently for each neuron from a Gaussian distribution $\mathcal{N}(1, 0.0025^2)$. In section 3.3, the task-specific input was added to the noisy one.

For the connectivity matrix $\mathbf{J} = \mathbf{J}_{\text{s}} + \mathbf{J}_{\text{h}}$, the noisy part was drawn independently as $(\mathbf{J}_{\text{h}})_{ij} \sim \mathcal{N}(0, \sigma^2/N_{\text{attr}})$, with $N_{\text{attr}} = 2500$ and $\sigma = 2$ in all experiments except for section 3.3, where we used $\sigma = 6$ for more robust trajectories. The symmetric part arranged the neurons on a 2D grid, such that every neuron $i$ had its coordinates $x_i$ and $y_i$ ranging from 0 to 49. The connectivity led to mutual excitation of nearby neurons and inhibition of the distant ones,
$$(\mathbf{J}_{\text{s}})_{ij} = -0.375 + \frac{1}{\sqrt{2\pi}} \exp\left(-d(i, j)^2/2\right), \tag{7}$$
$$d(i, j) = \frac{\pi}{L}\sqrt{(\min(|x_i - x_j|, L - |x_i - x_j|))^2 + (\min(|y_i - y_j|, L - |y_i - y_j|))^2}, L = 50. \tag{8}$$

The bump center (used in fig. 3B and D) corresponded to the mean of the activity on the torus. Denoting activity of each neuron as $r(x, y)$, the center on the $x$ axis was calculated as
$$x_{\text{center}} = \frac{L}{2\pi}\text{angle}\left(\sum_{n=0}^{L-1}\sum_{m=0}^{L-1}\frac{1}{L}r(x_n, y_m)e^{2\pi in/L}\right), \ L = 50, \tag{9}$$
where "angle" computes the counterclockwise angle of a complex variable (ranging from 0 to $2\pi$).

**Reservoir network parameters.** The time constant was $\tau = 50$ ms, and total coupling strength was $\lambda = 1.5$. The readout weights $\mathbf{W}_{\text{ro}}$ were initialized to zero. The feedback weights were drawn independently from a uniform distribution as $(\mathbf{W}_{\text{fb}})_{ij} \sim \mathcal{U}(-1, 1)$. Both the recurrent connections and the weights from the attractor to the reservoir were drawn independently as $(\mathbf{W}_{\text{rec}})_{ij}, (\mathbf{W}_{\text{attr}})_{ij} \sim \mathcal{N}(0, 1/pN_{res}) \cdot \text{Be}(p)$, with $p = 0.1$, $N_{res} = 1000$, and Be being the Bernoulli distribution. A new reservoir, and thus $\mathbf{W}_{\text{rec}}$, was sampled for each new test function. The matrix $\mathbf{W}_{\text{rec}}$ was the same for all tasks except the last one in section 3.3.

State noise $\boldsymbol{\xi}$ and exploratory noise $\boldsymbol{\eta}$ were generated independently from the uniform distribution as $\xi_i \sim \mathcal{U}(-0.05, 0.05)$, $\eta_i \sim \mathcal{U}(-0.5, 0.5)$. When attractor was present, the reservoir neurons also received weak independent noise drawn from $\mathcal{N}(0, 0.0025)$.

**Learning rule.** Low-pass filtering was done as
$$\bar{x}(t + dt) = \bar{x}(t) + dt\left(x(t) - \bar{x}(t)\right)/\tau_f, \ \tau_f = 5 \text{ ms}, \ \bar{x}(0) = 0. \tag{10}$$

The learning rate $\eta(t)$ was computed as $\eta(t) = \eta_0/(1 + t/\tau_l)$ ($\eta_0 = 5 \cdot 10^{-4}$, $\tau_l = 2 \cdot 10^4$ ms) and held at $\eta_0$ in section 3.2 to make conclusions independent of the decay.

