# OpenReview forum: "Working memory facilitates reward-modulated Hebbian learning in recurrent neural networks"
_NeurIPS.cc/2019/Workshop/Neuro_AI — Real Neurons & Hidden Units @ NeurIPS 2019 Poster_

### Official Review · AnonReviewer3 · 2019-09-26
**Interesting study with nice result showcasing biologically plausible learning rule for temporal sequences**

**Clarity:** 4

**Comment:**

As mentioned in technical rigor section, more rigorous investigation of the role of attractor for successful learning rule would be great addition to the paper.

**Category:**

Common question to both AI & Neuro

**Clarity Comment:**

The paper has few typos but is overall well written and easy to follow

**Evaluation:**

4: Very good

**Importance:**

3: Important

**Importance Comment:**

The ideas and results presented by the authors are novel and impressive.  Authors combined two biologically plausible concepts to construct their learning model: reward modulated Hebbian learning rule and working memory. It is an impressive fit that the model manages to achieve near-optimal performance as FORCE and also computationally cheap.

**Intersection:**

4: High

**Intersection Comment:**

The subject of network learning rule for complex temporal sequences is an important topic for both AI and neuroscience communities. The concepts discussed in the paper are good examples of biologically plausible interpretation of such learning rules, which appeals to both communities.

**Rigor Comment:**

While the results from the proposed model are impressive, to me I wish there were more in depth investigations on how bump attractor network is influencing the reservoir dynamics, and thus the convergence to the target signal with hebbian learning rule.

For instance, the authors states in Introduction "We propose stabilizing the reservoir activity by combining it with a continuous attractor" and "...feeding an abstract oscillatory input or a temporal backbone signal to the reservoir in order to overcome structural instabilities of the FORCE", but I am unable to find satisfactory explanations for these statements in the paper. According to the original paper by Sussillo and Abbot, initial chaotic state of the reservoir actually improved the training performance for particular choice of parameter 'g.' Does chaotic nature of the reservoir only inhibit model's learning capability? Could there be some choices of chaos parameter that actually helps learning? It would be interesting to look into these questions to understand better about the role of chaos, and inputs from attractor for successful learning.

**Technical Rigor:**

3: Convincing

---

> ### Author Response · Authors · 2019-10-22
> **Reply to the review**
>
> Thank you for the review!
>
> As we replied to Reviewer 2, theoretical analysis of the architecture would indeed be an interesting next step. Regarding our intuition see the earlier paragraph in response to reviewer 2.

---

### Official Review · AnonReviewer2 · 2019-09-27
**latent bump attractor network improves performance of RNN with local learning rule**

**Clarity:** 3

**Comment:**

The overall contribution of the paper is significant (in the sense of noticeable), but rather incremental.
* The biological plausibility of working memory implemented by bump attractors generated by 2500 firing rate unit (which usually represent large populations of spiking neurons) is questionable.
* It is not clear how "u" can be interpreted as a membrane potential when the entire network operates on the level of firing rates.
* It is not clear how delays impede the suppression of chaos
* It is not clear how this network would perform on other typical reservoir-computing tasks, e.g. the Romo task and how the performance improvements relate to "hints" given to the network during training (e.g. in full-FORCE).
* The performance should be compared to other learning algorithms for training RNN, especially those striving for  biological plausibility, e.g. feedback-alignment, Local online learning in recurrent networks with random feedback (RFLO) Murray 2019) and Eprop (Guillaume Bellec, Franz Scherr, Elias Hajek, Darjan Salaj, Robert Legenstein, Wolfgang Maass, 2019).
* Despite some shortcomings in the depth of the analysis, the paper altogether "good", especially the publication of the accompanying code is exemplary and enhances both understandability and reproducibility.

**Category:**

Common question to both AI & Neuro

**Clarity Comment:**

The problem is clearly explained. The details of the implementation are not described (time-step, adaptation parameter, gain parameter lambda (often called g for RNN)) but available in the accompanying code. The results are states clearly.

**Evaluation:**

3: Good

**Importance:**

3: Important

**Importance Comment:**

This work extends earlier work on reward-modulated Hebbian plasticity in RNN by a latent bump attractor network, which helps the network to bridge long timespans. The work provides no in-depth analysis of the mechanisms underlying the improved performance. Overall, it seems a small improvement compared to previous work. The author(s) provide code, which makes the paper completely reproducible.

**Intersection:**

3: Medium

**Intersection Comment:**

Learning long-term dependencies is challenging in RNN both in machine learning and in neuroscience because it requires bridging the time-scale from single neuron interactions (milliseconds) to the duration of tasks (seconds). While in the AI field, this is these days usually being addressed by gated units, the solution proposed here aims to achieve this with biologically plausible local learning rules in combination with a latent bump attractor. The proposed solution seems to my knowledge to be novel in the reservoir computing community, it has probably only limited relevance to the AI community (because they would just use gated units) and but seems moderately relevant for the neuroscience community.

**Rigor Comment:**

Generally, the results seem plausible and sound. However, the putative mechanism behind the improved performance (slow dynamics of bump attractor bridging the timescale from short Hebbian plasticity to long timespans of the task), is only hypothesized but not actually studied. Also, a mechanistic understanding of how chaos is being suppressed during training is missing. The paper compares the novel learning algorithm on one single toy example to is predecessor, so it is difficult to see how its performance compares with alternative approaches. A huge bonus is that the author(s) provide code, which makes the paper completely reproducible.

**Technical Rigor:**

3: Convincing

---

> ### Author Response · Authors · 2019-10-22
> **Reply to the review**
>
> Thank you for your review!
>
> In the following we would like to address your comments on the work:
>
> 1. Lack of mechanistic understanding (Rigor comment)
>
> Indeed, we only experimented with how input from a dynamic attractor shapes activity of the reservoir on few example tasks. It would be interesting to provide a theoretical explanation of why the proposed scheme improves reward-modulated Hebbian learning, however, this is left for future work. Our intuition is the following: the slow input from the attractor network serves as a (time-dependent) bias that puts the reservoir network into different states as the task progresses on the slow time scale. At each segment of the task (i.e., for a given bias) the fast dynamics of the reservoir network works in the same way as in other reservoir tasks. So in our view, our approach is akin to shifting continuously through a family of reservoir networks (defined by different input bias) while the readout stays fixed throughout.
>
> 2. Test on a single example only (Rigor comment and Final comment)
>
> Although we only tested the network on two types of tasks (1D periodic functions and 2D drawings), the first task actually included 100 different (randomly sampled) functions, 50 for the 1s task and 50 for the 10s one, as opposed to a few hand-picked test functions that were used in the preceding work cited in our paper. Our approach showed very little variance in performance across the different functions. It would be interesting to also test the model on the Romo task as specific biological example. This is, however, left for future work.
>
> 3. Simulation details (Clarity comment)
>
> For the final camera-ready version of the paper, we have now included the simulations details in the appendix.
>
> 4. Attractor network, rate units, and membrane potential (Final Comment)
>
> Yes, we work with firing rate units in an attractor network of a given size. In principal, the size of the attractor network can be increased or reduced. Combining it with external cues can also be helpful, so as to reuse the same trajectory for multiple target functions. In rate networks, the variable u indeed only distantly resembles the membrane potential; we prefer the wording membrane potential to the abstract notion  ‘internal state of the neuron’.  Note that attractor networks can also be formulated with spiking neurons with adaptation and short-term depression, and then the membrane potential is well defined see e.g., Seeholzer et al. (2019), PLOS Comput. Biol. 15:e1006928. doi: 10.1371/journal.pcbi.1006928

---

### Decision · Program_Chairs · 2019-10-02

Accept (Poster)